# Expression Profile of Fibroblast Growth Factor Receptors, Keratinocyte Differentiation Markers, and Epithelial Mesenchymal Transition-Related Genes in Actinic Keratosis: A Possible Predictive Factor for Malignant Progression?

**DOI:** 10.3390/biology10040331

**Published:** 2021-04-15

**Authors:** Flavia Persechino, Danilo Ranieri, Luisa Guttieri, Monica Nanni, Maria Rosaria Torrisi, Francesca Belleudi

**Affiliations:** 1Department of Clinical and Molecular Medicine, Sapienza University of Rome, 00189 Rome, Italy; flavia.persechino@uniroma1.it (F.P.); danilo.ranieri@uniroma1.it (D.R.); luisa.guttieri@uniroma1.it (L.G.); mara.torrisi@uniroma1.it (M.R.T.); 2Tissue Biology Research Unit, Department of Surgery, University Children’s Hospital, CH–8032 Zurich, Switzerland; monica.nanni@kispi.uzh.ch; 3S. Andrea University Hospital, 00100 Rome, Italy

**Keywords:** FGFR, FGFR2, EMT, keratinocyte differentiation, actinic keratosis

## Abstract

**Simple Summary:**

In this work, we checked the modulation of Fibroblast Growth Factor Receptors (FGFRs) along with differentiation-related and epithelial-to-mesenchymal transition (EMT)-related markers to identify expression profiles that could be predictive for actinic keratosis (AK) progression through the “differentiated” pathway. We found that the downregulation of the analyzed differentiation markers, but not the modulation of the EMT-related markers, correlated with the canonical progression of AK. In addition, the observed modulation of FGFR2 mesenchymal/epithelial isoforms compatible with FGFR2 isoform switch, as well as the upregulation of FGFR4 suggested their correlation with early steps of AK pathogenesis. In contrast, the increase of mesenchymal FGFR3c isoform expression appeared to suggest that this event correlated with late steps of AK progression. In addition, the strong modulation of filaggrin (FIL), Snail1, as well as of FGFR2c, FGFR4, and their ligand Fibroblast Growth Factor 2 (FGF2), observed in some of the keratinocytic intraepithelial neoplasia grade I (KIN I) samples, may indicate that they could be molecular markers predictive for those KIN I lesions destined to a direct progression to squamous cell carcinoma (SCC) through the “differentiated” pathway.

**Abstract:**

Actinic keratosis (AK) is the ultra violet (UV)-induced preneoplastic skin lesion clinically classified in low (KIN I), intermediate (KIN II), and high (KIN III) grade lesions. In this work we analyzed the expression of Fibroblast Growth Factor Receptors (FGFRs), as well as of keratinocyte differentiation and epithelial-to-mesenchymal transition (EMT)-related markers in differentially graded AK lesions, in order to identify specific expression profiles that could be predictive for direct progression of some KIN I lesions towards squamous cell carcinoma (SCC). Our molecular analysis showed that the keratinocyte differentiation markers keratin 1 (K1), desmoglein-1 (DSG1), and filaggrin (FIL) were progressively downregulated in KIN I, II, and III lesions, while the modulation of epithelial/mesenchymal markers and the induction of the transcription factors Snail1 and Zinc finger E-box-binding homeobox 1 (ZEB1) compatible with pathological EMT, even if observable, did not appear to correlate with AK progression. Concerning FGFRs, a modulation of epithelial isoform of FGFR2 (FGFR2b) and the mesenchymal FGFR2c isoform compatible with an FGFR2 isoform switch, as well as FGFR4 upregulation were observed starting from KIN I lesions, suggesting that they could be events involved in early steps of AK pathogenesis. In contrast, the increase of FGFR3c expression, mainly appreciable in KIN II and KIN III lesions, suggested a correlation with AK late progression. Interestingly, the strong modulation of FIL, Snail1, as well as of FGFR2c, FGFR4, and of their ligand FGF2, observed in some of the KIN I samples, may indicate that they could be molecular markers predictive for those low graded lesions destined to a direct progression to SCC. In conclusion, our data point on the identification of molecular markers predictive for AK rapid progression through the “differentiated” pathway. Our results also represent an important step that, in future, will help to clarify the molecular mechanisms underlying FGFR signaling deregulation in epithelial tissues during the switch from the pre-neoplastic to the oncogenic malignant phenotype.

## 1. Introduction

Actinic keratosis (AK) is a UV-induced preneoplastic skin lesion characterized by cutaneous dysplasia of epidermal keratinocytes, with a progressive trend (6% of AKs) towards squamous cell carcinoma (SCC) [1,2]. In early lesions, atypical keratinocytes (altered in size, shape, and organization) are present in the basal layer and progressively extend throughout the entire epidermis during progression. Keratinocyte differentiation is defective and results in parakeratosis alternating with hyperkeratosis [1,2], while the downregulation of several differentiation markers [3], as well as high expression of basal markers keratin 14 (K14) and K17 in the suprabasal layers [4,5] have been proposed as a prognostic factor for the progression to SCC.

Due to its similarity with cervical intraepithelial neoplasia in terms of biological behavior and progression, a clinical classification of AK in keratinocytic intraepithelial neoplasia (KIN) of three histo-morphological grades (‘low grade’ or KIN I, ‘intermediate grade’ or KIN II, and ‘high grade’ or KIN III) has been proposed [6,7]. Alongside the “canonical” progression, a so-called “differentiated” pathway has also been proposed for some early KIN I lesions, able to directly evolve in aggressive SCCs [6,8] and presenting the epithelial-mesenchymal transition (EMT) as a typical feature [9].

Concerning Fibroblast Growth Factor Receptors (FGFRs), several findings strongly suggested that the dysregulation of FGFR expression and signaling could also contribute to AK pathogenesis [10,11,12]. This possibility is also sustained by our recent observations, demonstrating that, while the epithelial isoform of FGFR2 (FGFR2b) is directly implicated in keratinocyte differentiation [13], the altered FGFR2 isoform switching and the consequent aberrant expression of the mesenchymal FGFR2c isoform in human keratinocytes induce changes in FGF specificity, leading to impairment of differentiation [14] and induction of EMT and tumorigenic features [15,16]. Even if our findings are consistent with the opposite oncogenic and tumor-suppressive roles previously proposed for FGFR2c and FGFR2b, respectively [17,18,19], the specific function of each FGFR2 isoform in carcinogenesis, and in particular their role during the progression of pre-malignant lesions, such as AK, remains to be further clarified.

In this paper, we aimed to assess the expression pattern of FGFRs and of the ligand FGF2, as well as that of early and terminal differentiation markers and EMT-related genes in differently graded AK lesions (KIN I, KIN II; KIN III), in order to clarify if specific gene expression profiles could be a useful prognostic tool to identify those early AK lesions with highest probability of direct progression towards SCC.

## 2. Materials and Methods

### 2.1. Institutional Review Board Statement

All subjects gave their informed consent for inclusion before they participated in the study. The study was conducted in accordance with the Declaration of Helsinki, and the protocol was approved by The Institutional Review Board of ‘‘Sapienza’’ University and Sant’Andrea Hospital (protocol n°176/2011).

### 2.2. Histological Samples and Primary Cell Cultures

Histological samples were obtained from actinic keratosis (AK) lesions with Histo-morphological evaluation consistent with low grade (KIN I; *n* = 6) intermediate grade (KIN II; *n* = 3) and high grade (KIN III; *n* = 2) keratinocytic intraepithelial neoplasia and from the corresponding perilesional area (PL), were obtained from patients attending the Dermatology Unit of the Sant’Andrea Hospital of Rome. All patients were extensively informed and their consent for the investigation was collected in written form in accordance with guidelines approved by the management of the Sant’Andrea. The samples, identified by an alphanumeric code (e.g., p1AK, p1PL), were separated in epidermal and dermal portion by digestion in 12 U/mL of dispase in Hank’s buffered salt solution, before total RNA extraction. Primary cultures of human fibroblasts derived from the dermal portion of each sample, were isolated and cultured as previously described [20].

### 2.3. Immunofluorescence

Primary cultures of Human Fibroblasts (HFs) were grown on coverslips, fixed with 4% paraformaldehyde in PBS for 30 min at 25 °C followed by treatment with 0.1 M glycine for 20 min at 25 °C and with 0.1% Triton X-100 for an additional 5 min at 25 °C to allow permeabilization. Cells were then incubated with the primary polyclonal antibodies anti-vimentin (1:50 in PBS; Dako, Glostrup, Denmark) for 1 h at 25 °C and then with a FITC-conjugated goat anti-rabbit IgG (1:400 in PBS; Cappel Research Products, Durham, NC, USA) for 30 min at 25 °C. Nuclei were stained with 4′,6-diamidino-2-phenylindole (DAPI) (1:1000 in PBS; Sigma-Aldrich, Saint Louis, MO, USA). Coverslips were finally mounted with mowiol solution (Sigma) and analyzed using a conventional fluorescence microscope.

### 2.4. Primers

Oligonucleotide primers for target genes and for the housekeeping gene were chosen with the assistance of the Oligo 5.0 computer program (National Biosciences, Plymouth, MN) or the online tool Primer-BLAST [21] and purchased from Invitrogen (Invitrogen, Carlsbad, CA, USA). The following primers were used: for FGFR1b target gene 5′-CGGGGATTAATAGCTCGGATG-3′ (sense), 5′-GCACAGGTCTGGTGACAGTGA-3′ (antisense); for FGFR1c target gene 5′-TGGGAGCATTAACCACACCTACC-3′ (sense), 5′-GCACCTCCATTTCCTTGTCG-3′ (antisense); for FGFR2b/KGFR target gene: 5′-CGTGGAAAAGAACGGCAGTAAATA-3′ (sense), 5′-GAACTATTTATCCCCGAGTGCTTG-3′ (antisense); for FGFR2c target gene: 5′-TGAGGACGCTGGGGAATATACG-3′ (sense), 5′-TAGTCTGGGGAAGCTGTAATCTCCT-3′ (antisense); for FGFR3b target gene 5′-TGCTGAATGCCTCCCACG-3′ (sense), 5′-CGAGGATGGAGCGTCTGTC-3′ (antisense); for FGFR3c target gene 5′-CGCCCTACGTCACTGTACTCAA-3′ (sense), 5′-GTGACATTGTGCAAGGACAGAAC-3′ (antisense); for FGFR4 target gene 5′-CTGTGGCCGTCAAGATGCTCAA-3′ (sense), 5′-ATGTTCTTGTGTCGGCCGATCA-3′ (antisense); for DSG1 target gene 5′-GTGGGAGAAAGAAAAAGAACAGAGAAG-3′ (sense), 5′-CTACCACCACCAGAAAAATGAACAG-3′ (antisense); for FIL target gene 5′-TTTCGGCAAATCCTGAAGAATCC-3′ (sense), 5′-CTTGTTGTGGTCTATATCCAAGTGATC-3′ (antisense); for K1 target gene 5′-AGCACAAGCCACACCACCATC-3′ (sense), 5′-CGCCACCTCCAGAACCATAGC-3′ (antisense); for Snail1 target gene: 5′-GCTGCAGGACTCTAATCCAGA-3′ (sense), 5′-ATCTCCGGAGGTGGGATG-3′ (antisense); for ZEB1 target gene 5’-GGGAGGAGCAGTGAAAGAGA-3’ (sense), 5’-TTTCTTGCCCTTCCTTTCTG-3’ (antisense), for vimentin target 5′-AATCCAAGTTTGCTGACCTCTCTG-3′ (sense), 5′-TCATTGGTTCCTTTAAGGGCATCC-3′ (antisense), for E-cadherin target 5′-TGGAGGAATTCTTGCTTTGC-3′ (sense), 5′-CGCTCTCCTCCGAAGAAAC-3′ (antisense); for FGF2 target 5′- ATGGCAGCCGGGAGCATCACCCACG-3′ (sense), 5′- TCAGCTCTTCGCAGACATTGGAAG-3′ (antisense); for the 18S rRNA housekeeping gene 5′-CGAGCCGCCTGGATACC-3′ (sense), and 5′-CATGGCCTCAGTTCCGAAAA-3′ (antisense). For each primer pair, no-template control and no-reverse-transcriptase control (RT negative) assays were performed, which produced negligible signals.

### 2.5. RNA Extraction and cDNA Synthesis

RNA was extracted using the TRIzol method (Invitrogen, Carlsbad, CA, USA) according to manufacturer’s instructions and eluted with 0.1% diethylpyrocarbonate (DEPC)-treated water. Each sample was treated with DNAase I (Invitrogen). Total RNA concentration was quantitated by spectrophotometry; 1 μg of total RNA was used to reverse transcription using iScriptTM cDNA synthesis kit (Bio-Rad Laboratories, Hercules, CA, USA) according to manufacturer’s instructions.

### 2.6. PCR Amplification and Real-Time Quantitation

Real-Time PCR was performed using the iCycler Real-Time Detection System (iQ5 Bio-Rad) with optimized PCR conditions. The reaction was carried out in 96-well plate using iQ SYBR Green Supermix (Bio-Rad) adding forward and reverse primers for each gene and 1 μL of diluted template cDNA to a final reaction volume of 15 μL. All assays included a negative control and were replicated three times. The thermal cycling program was performed as described [14]. Real-time quantitation was performed with the help of the iCycler IQ optical system software version 3.0a (Bio-Rad), according to the manufacturer’s manual. Threshold cycle values were calculated using the Pfaffl method and specificity of PCR products was verified by melting curve analysis. The relative expression of the housekeeping gene was used for standardizing the reaction.

Values of the target genes K1, DSG-1, FIL, E-cadherin, vimentin, Snail1, ZEB1, FGFR1b, FGFR2b, FGFR3b, and FGFR4 were normalized in respect to the mean of values obtained by all epidermal perilesional samples. Values of the target genes FGFR1c, FGFR2c, FGFR3c were normalized in respect to the value of primary culture of HFs isolated from a dermal perilesional sample. Finally, the values of FGF2 target gene were normalized in respect to the mean of values obtained by all dermal perilesional samples.

### 2.7. Statistical Analysis

For statistical analysis of relative expression rates of AK versus PL samples, the Wilcoxon rank-sum test for dependent samples was applied. All the results were expressed as the mean ± standard deviation (SD), and significance level was defined as *p* < 0.05.

## 3. Results

### 3.1. AK Lesions Display Modulation of Differentiation and of EMT-Related Markers

First, we focused our attention on keratinocyte differentiation markers, comparing their expression levels in the epidermal portion of lesional KIN I, KIN II, and KIN III samples with that detected in the same portion of corresponding perilesional controls. As a first step, all lesional and perilesional samples were separated in epidermal and dermal portion by digestion in dispase before total RNA extraction, as reported in Materials and Methods. Real time RT-PCR analysis performed using RNAs extracted from the epidermal portions, showed a reduction trend of the expression of the early differentiation marker K1 in all the three groups (Figure 1A). The effect on differentiation was further assessed, analyzing the impact on the expression of DSG1, a desmosomal component expressed starting from the suprabasal layer [22] and directly involved in the initiation of keratinocyte early differentiation [23,24]. Results showed a reduction trend of DSG1 expression, that became progressively more evident moving from KIN I and II to KIN III group (Figure 1A). Thus, the pathological condition of AK appears to interfere with the expression of both K1 and DSG1, confirming the previously described impact on the onset of keratinocyte differentiation [3]. In order to check if AK could also impact on later stages of differentiation, we analyzed the expression of a specific terminal differentiation marker, FIL. Molecular analysis revealed that the downregulation of FIL, even if detectable in all AK samples, appeared particularly evident in some of the KIN I samples, compared to their corresponding perilesional controls (see in particular samples p4PL/AK, p6PL/AK, and p11PL/AK) (Figure 1B). A further validation of the results concerning the differentiation markers was obtained comparing mean values of all KIN I samples versus the mean values of all the corresponding perilesional controls and performing statistical analysis by Wilcoxon test for dependent samples. Results showed a decreasing trend of K1 and DSG1 and a significant downregulation of FIL (Figure 1C). A parallel analysis performed comparing the mean values of the entire cohort of all the AK lesions versus their perilesional controls, performed using the same statistical approach, displayed a significant downregulation of all the three differentiation markers (Figure 1D). Our results appeared to indicate that while the impairment of the early differentiation is an event that gradually accompanies KIN I, KIN II, KIN III progression, the impairment of the terminal differentiation marker FIL, even if it affects all lesions, seems to be a phenomenon particularly accentuated in some of the early KIN I lesions. This last observation suggested that FIL could be one of the specific markers whose modulation could be useful to identify KIN I lesions destined for a rapid malignant progression.

Since EMT profile is a typical feature of aggressive SCCs directly developing from KIN I [9], a detailed analysis of EMT markers in the epidermal portions of all grades of AK lesions, with particular attention to KIN I samples, could help to clarify the AK-related gene expression profile with a higher probability to evolve towards SCC. To this aim, we first focused on the modulation of epithelial/mesenchymal markers and we found that the downregulation of the epithelial marker E-cadherin and the upregulation of the mesenchymal marker vimentin appeared detectable throughout all the three groups of AK samples (Figure 2A). Interestingly, the slight expression of vimentin, detectable also in perilesional controls (Figure 2A), but not in unrelated epidermal samples from healthy skin (data not shown), suggested a possible beginning of subversion of the epithelial/mesenchymal marker expression in the area adjacent to AK lesions. These observations are consistent with the widely proposed concept of “cancerization field” [25]. It is well known that changes in phenotypic features during EMT are the results of a complex gene reprogramming driven by different transcription factors. ZEB1, in particular, plays a central role in the repression of several epithelial markers, including E-cadherin [26,27,28], while Snail1 is the widely recognized master for EMT transcription factor [26,27], mainly associated with the initiation of the process [29]. Therefore, the expression profile of these key transcription factors was also investigated, in order to describe their possible dysregulation in differently graded AK lesions. Real-time RT-PCR showed that an increase of both ZEB1 and Snail expression was evident in all KIN subgroups, compared to the corresponding perilesional controls. Moreover, Snail1 displayed a more marked inter- and intra-sample variability, compared to ZEB1 (Figure 2B). Interestingly, the high increase of Snail1 observed in some of the KIN I samples (see p1PL/AK and p9 PL/AK), appear to indicate that also this transcription factor might be a marker that could characterize those KIN I lesions destined to a rapid malignant progression. The comparation of mean values of all KIN I samples versus their controls, and statistical analysis by Wilcoxon test confirmed the opposite trend of E-cadherin and vimentin genes (Figure 2C), as well as the induction of both EMT-related transcription factors ZEB1 and Snail1 (Figure 2C). A parallel analysis performed comparing the mean values of the entire cohort of AK lesions versus their corresponding perilesional controls, revealed comparable trends, which became significant for all the analyzed genes (Figure 2D).

### 3.2. Differentially Graded AK Lesions Show Differential Modulation of FGFRs

The accumulating observations suggesting that FGF/FGFR signaling could be crucial for AK progression towards SCC [10,11,12] brought us to focus on the FGFR expression profile. Moreover, our recent evidence of a role played by the FGFR2 isoform switch and consequent aberrant expression of the mesenchymal FGFR2c isoform in EMT induction and impairment of differentiation in human keratinocytes [14,15,16], encouraged us to pay particular attention to FGFR2 isoforms in our investigation. Real time RT-PCR performed using RNAs from the epidermal portion of all lesional and perilesional samples, showed a high inter and intra-sample variability of the expression of both epithelial FGFR1b, and FGFR3b isoforms (Figure 3A,C). Concerning their mesenchymal counterparts, while FGFR1c expression was undetectable in all samples (Figure 3A), the expression of FGFR3c, which was very low in KIN I samples, excluding (p1AK sample) (Figure 3C), appeared mostly increased in KIN II and KIN III lesions, compared to their corresponding perilesional controls (Figure 3C). Focusing on FGFR2 isoforms, we found that the expression of the mesenchymal FGFR2c isoform resulted increased in AK lesions already stating from KIN I samples (Figure 3B) and this induction was accompanied by a reduction of epithelial FGFR2b isoform (Figure 3B) suggesting an isoform switch event. Similar to what was observed for FGFR2c, also FGFR4 appears increased in all AK lesions, starting from KIN I samples (Figure 3D). It is worth noting that a high intra-sample increase of both FGFR2c and FGFR4 was observed in KIN I sample p1PL/AK, which similarly displayed a strong induction of FGFR3c, as well as of the Snail1 gene. These results suggested that the FGFR2c/FGFR3c/FGFR4/Snail1 expression profile could be indicative of KIN I lesions with a malignant tendency. Moreover, as previously observed for vimentin, the mesenchymal FGFR2c isoform, as well as FGFR4, which are not normally expressed in keratinocytes [14], appeared detectable also in perilesional controls (Figure 3B,D), suggesting a possible initiation of FGF/FGFR axis subversion in the area adjacent to AK lesions compatible with the establishment of a “cancerization field” [25]. The analysis of data from all KIN I sample compared their perilesional controls, confirmed the opposite trend of FGFR2b and FGFR2c genes (Figure 3E), as well as the induction of FGFR4 (Figure 3E), which became significant when the entire cohort of AK lesions were statistically analyzed by Wilcoxon test (Figure 3F).

### 3.3. Independently from the Progression Grade, AK Lesions Show Increased Expression of FGF2

In concert with the potential impairment of FGFR expression in keratinocytes, we therefore wanted to further investigate whether a possible deregulation of fibroblast functions, and in particular of their ability to release FGFs, could contribute to the establishment of a dysregulated FGF/FGFR paracrine axis in AK lesions. In order to further explore this, we first focused on the phenotype displayed by primary culture of fibroblasts isolated from the dermal portion of different KIN lesions, comparing it with the phenotype displayed by fibroblasts derived from corresponding perilesional controls. To confirm the mesenchymal profile and the purity of the primary cultures, the assessment of the expression of vimentin, a component of the intermediate filaments widely used as fibroblast-specific marker, was performed by immunofluorescence. The results showed that, independently from AK grades or corresponding perilesional origin, all the analyzed cultures were highly positive for the staining with anti-vimentin antibodies (Figure 4). As expected, the fluorescent signal displayed a distribution compatible with the structure and localization of cytoplasmic bundles of vimentin filaments (Figure 4). In parallel, all cultures appeared negative to the staining with anti-pan-cytokeratin antibodies (data not shown), confirming the absence of contaminant epithelial cells and the specific mesenchymal phenotype of all the isolated primary cultures. In addition, primary cultures from all AK samples did not show relevant differences in term of growth mode and cell shape, if compared to the cultures isolated from the corresponding perilesional controls. Then, the ability of dermal fibroblasts to express and possibly release FGF2, an FGF family member activating several FGFRs, including FGFR4 and FGFR2c, but not its epithelial counterpart FGFR2b, was investigated by molecular approaches. Real time RT-PCR, performed using RNA extracts from the dermal portion of both AK and perilesional biopsies, showed that, even if FGF2 mRNA transcript modulation appeared highly variable (Figure 5A), the increasing trend observed when all samples of KIN I lesions were compared to their controls (Figure 5B) became significant when the entire cohort of all AK lesions were considered (Figure 5C).

Therefore, although primary cultures of fibroblasts from AK samples, independently from KIN grade, did not display relevant differences in terms of cell morphology and growth mode if compared to their perilesional counterpart, the significant increase in the FGF2 mRNA transcripts detected in the dermal portion of all AK lesions, compared to the corresponding perilesional controls, indicated an increased ability of all AK fibroblasts to release this growth factor.

Interestingly, as observed for FGFR2c, FGFR4, and Snail 1, the same KIN I sample p1AK also showed a strong increase of FGF2 expression compared to its corresponding perilesional control (p1PL) suggesting that all these proteins could belong to a pattern of markers whose expression profile could help to characterize KIN I lesion destined to a rapid and direct progression to SCC.

## 4. Discussion

The progression of the pre-neoplastic AK lesions towards SCC is accompanied by a clear downregulation of several keratinocyte differentiation markers [3], while the appearance of an EMT phenotype is a typical feature of aggressive SCCs directly developing from early KIN I lesions [9]. Based on these observations, in the present study we first analyzed the expression of both differentiation-related and EMT-related genes, in different KIN I, II, and III samples, in order to establish if specific expression profiles could be predictive for AK malignant progression.

In addition, the observation that FGFs were upregulated in both AK keratinocytes and in cancer associated fibroblasts (CAFs) [11] suggested that the dysregulation of FGFs/FGFRs axis could contribute to AK pathogenesis. This possibility was also sustained by the identification of the FGF1/FGF2 inhibitor (Dobesilate) as a new effective therapeutic strategy for AK lesions [10,12]. Among FGFRs, FGFR2 could, play a crucial role, as suggested by our recent findings demonstrating that FGFR2 isoform switching and the aberrant expression of the mesenchymal isoform FGFR2c in human keratinocytes induce impairment of differentiation [14], EMT, and tumorigenic features [15,16]. In light of these observations, it is reasonable to suppose that, in pre-malignant AK lesions destined to a rapid aggressive progression, the altered splicing of FGFR2 could take place as an early event leading to the impairment of differentiation and EMT induction, both typical features possibly required for the progression of KIN I lesions directly to SCC. In agreement with this possibility, our previous reports demonstrated that UV irradiation, which is the main cause of AK, induces FGFR2b downregulation [30,31], which in turn causes the attenuation of survival signals in stratified keratinocytes [32]. In this scenario, the possibility that UV exposure could unbalance skin homeostasis via a more articulated mechanism also involving the FGFR2b/FGFR2c isoform switching cannot be excluded. Based on these speculations, in this work we also checked FGFR family expression profiles in our collection of differently graded AK samples, paying particular attention to FGFR2 isoforms.

Overall, the molecular analysis highlighted that: (i) the early differentiation markers K1 and DSG1 appeared progressively downregulated during KIN I, II, and III progression, suggesting that the impairment of the onset of differentiation could be an effect which begins to be manifested in early stages and continues to interest AK progression, while the downmodulation of the terminal differentiation marker FIL, particularly marked just in some of the KIN I lesions, suggested that FIL could be one of the markers whose repression could characterize KIN I lesions destined to a rapid malignant progression; (ii) a modulation of epithelial/mesenchymal markers and an induction of the EMT-related transcription factors Snail1 and ZEB1, compatible with pathological EMT, were observed in all the analyzed AK samples, without significant differences between KIN histo-morphological groups; (iii) a modulation of FGFR2b/FGFR2c isoforms, compatible with an FGFR2 isoform switch, as well as an upregulation of FGFR4, were detected starting from KIN I lesions, encouraging us to suppose that they could be key events in the early steps of AK pathogenesis, while the appearance of FGFR3c, mainly relevant in KIN II and KIN III lesions, suggested that it could be a later event correlated to AK progression; (iv) finally, the dermal portion of AK samples displayed increased FGF2 mRNA expression, suggesting enhanced ability of AK fibroblasts to release this growth factor and consequently to possibly contribute to aberrant epidermal/dermal paracrine loops based on FGF2/FGFR axis. Our results were consistent with the findings of several other groups, extensively highlighting and discussing the essential role played by the paracrine loops between tumor cells and cancer-associated fibroblasts (CAF) in several tumor contexts, including SCC [33,34]. In these complex cross-talks, the fibroblasts act by supplying several cytokines and growth factors, including FGF, essential for tumor development and progression.

It is worth noting that, particularly in one of the samples of the KIN I group, a strong induction of Snail1, as well as the upregulation of FGFR2c, FGFR4, and FGF2 was observed. Even if the number of samples analyzed for each AK subgroups was very small to draw definitive conclusions, this observation encouraged us to propose that these proteins could belong the pool of molecular markers, whose modulation could in future help to quickly intercept KIN I lesion destined to a rapid and direct progression towards SCC. The establishment of the expression profile of these molecular markers in AK biopsies could be, in future, a useful prognostic application for identifying those early lesions that need to be removed due to high probability of malignant evolution. Although the role of FGFR2c expression in the epithelial context begins to be clarified, the specific signaling network activated downstream this receptor and underlying its oncogenic outcome still remain to a large extent to be identified. Concerning this topic, we recently identified Protein Kinase Cε (PKCε)-mediated signaling as being mainly responsible for FGFR2c-mediated EMT and tumorigenic features in human keratinocytes [35,36]. Among the various transcription factors involved in the malignant progression and that can be therapeutically targeted [37], we found that PKCε acts downstream FGFR2c regulating Snail1, Fos-Related Antigen-1 (FRA1), and Signal Transducer and Activator of Transcription 3 (STAT3), which are induced in cascade [35]. Actually, the activation of PKCε-mediated signaling in consequence of FGFR2 isoform switch and the involvement of this pathway in FGFR2c-mediated epidermal carcinogenesis could explain the higher occurrence of SCC in AK lesions treated with ingenol mebutate, the most common topical therapy used in the last few years for AK treatment. In fact, the oncogenic effect proved for ingenol mebutate, which costed the recent suspension of the license by the European Medicines Agency (EMA) (DTB team, Drug and Therapeutics Bulletin, 2020), could be explained by the ability of this drug to strongly activate PKCs, including PKCε [38]. In light of the finding reported in the present work, it is reasonable to suppose that, in precocious AK lesions presenting FGFR2 isoforms switch, as well as enhanced secretion of FGF2, Ingenol Mebutate could act as amplifier for the PKCε-mediated oncogenic signaling (a signaling already triggered by the dysregulated FGF/FGFR paracrine axis), favoring the rapid progression of the KIN I lesions towards skin malignancies. Further investigations, including the assessment of the PKC expression profile in our AK sample collection, are in progress to elucidate this possibility.

## 5. Conclusions

Our findings showed that the downregulation of the differentiation markers analyzed, but not the modulation of EMT-related markers, correlated with AK progression. Moreover, a modulation of FGFR2b and FGFR2c compatible with an FGFR2 isoform switch, as well as FGFR4 upregulation were observed starting from KIN I lesions, suggesting that they could be events correlated to the early steps of AK pathogenesis. In contrast, the increase of FGFR3c expression, mainly appreciable in KIN II and KIN III lesions, suggested a correlation with the late steps of AK progression. Finally, the strong modulation of some of the proteins analyzed (FIL, Snail1, FGFR2c, FGFR4, and FGF2) in specific KIN I samples, suggested that they could be molecular markers predictive for AK progression through the “differentiated” pathway. Our study represents an advancement of knowledge on the molecular bases of FGF/FGFR axis deregulation, occurring in epithelial tissues during the progression from the pre-neoplastic to the oncogenic malignant phenotype.

## Figures and Tables

**Figure 1 biology-10-00331-f001:**
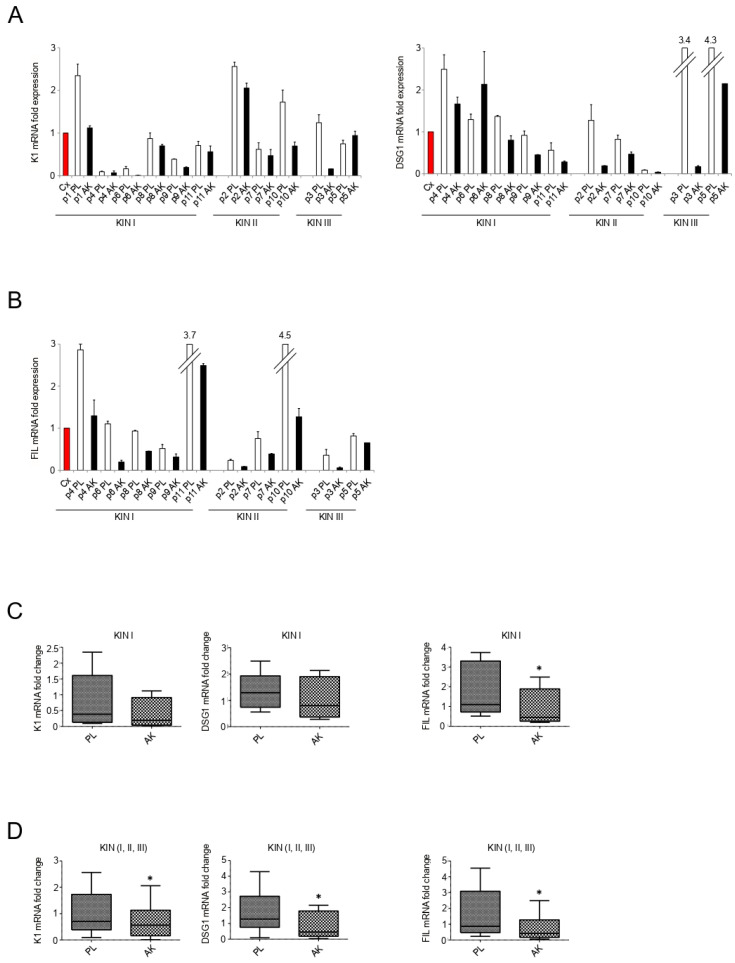
Expression of keratinocyte differentiation markers in the epidermal portion of KIN I, KIN II, and KIN III samples. K1 (**A**), DSG1 (**A**), and FIL (**B**) mRNA levels were evaluated by real-time RT-PCR in KIN I, KIN II, and KIN III samples (PL: perilesional samples, white bars; AK: lesional samples, black bars) and normalized in respect to the mean of all the perilesional values (Cx, red bar). Results are expressed as mean ± standard error (SE). Statistical analysis of relative expression rates of KIN I AK versus KINI PL samples (**C**) or KIN I + KIN II + KIN III AK versus KIN I + KIN II + KIN III PL samples (**D**) was performed using Wilcoxon rank-sum test and significance levels were defined as *p* < 0.05: * *p* < 0.05 vs. the corresponding PL samples.

**Figure 2 biology-10-00331-f002:**
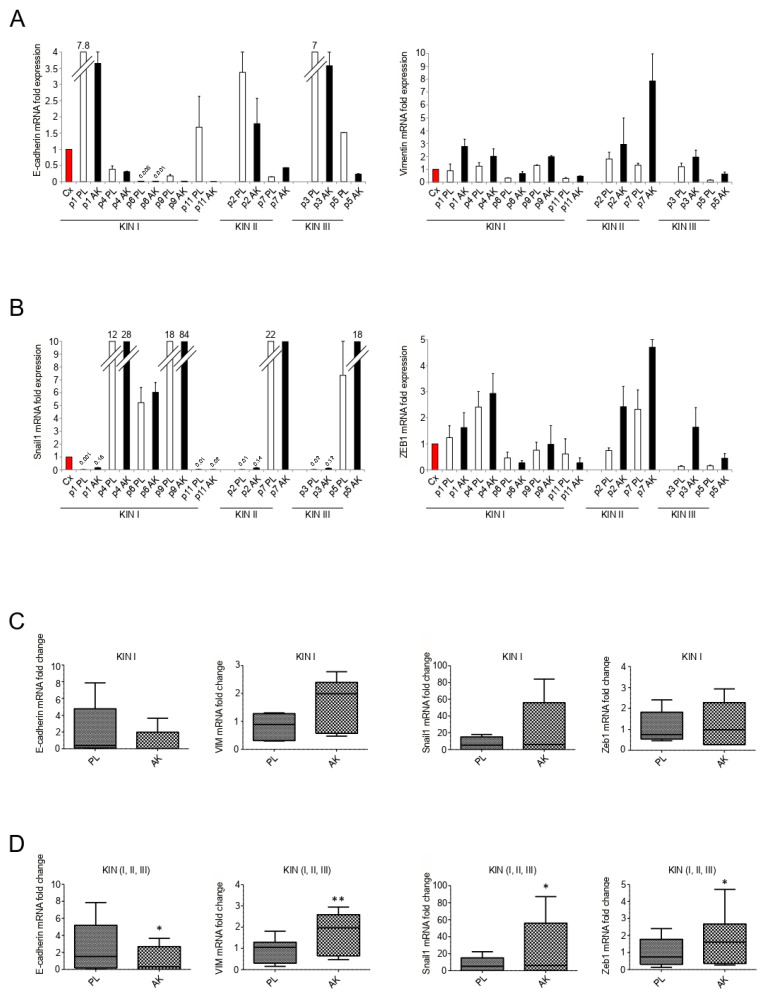
Expression of epithelial-to-mesenchymal transition (EMT)-related markers in the epidermal portion of KIN I, KIN II, and KIN III samples. E-cadherin (**A**), vimentin (**A**), Snail1, and ZEB1 (**B**) mRNA levels were evaluated by real-time RT-PCR (PL: perilesional samples, white bars; AK: lesional samples, black bars) and normalized with respect to the mean of all the perilesional values (Cx, red bar). Results are expressed as mean ± standard error (SE). Statistical analysis of relative expression rates of KIN I AK versus KINI PL samples (**C**) or of KIN I + KIN II + KIN III AK versus KIN I + KIN II + KIN III PL samples (**D**) was performed using Wilcoxon rank-sum test and significance levels were defined as *p* < 0.05: * *p* < 0.05, ** *p* < 0.001 vs. the corresponding PL samples.

**Figure 3 biology-10-00331-f003:**
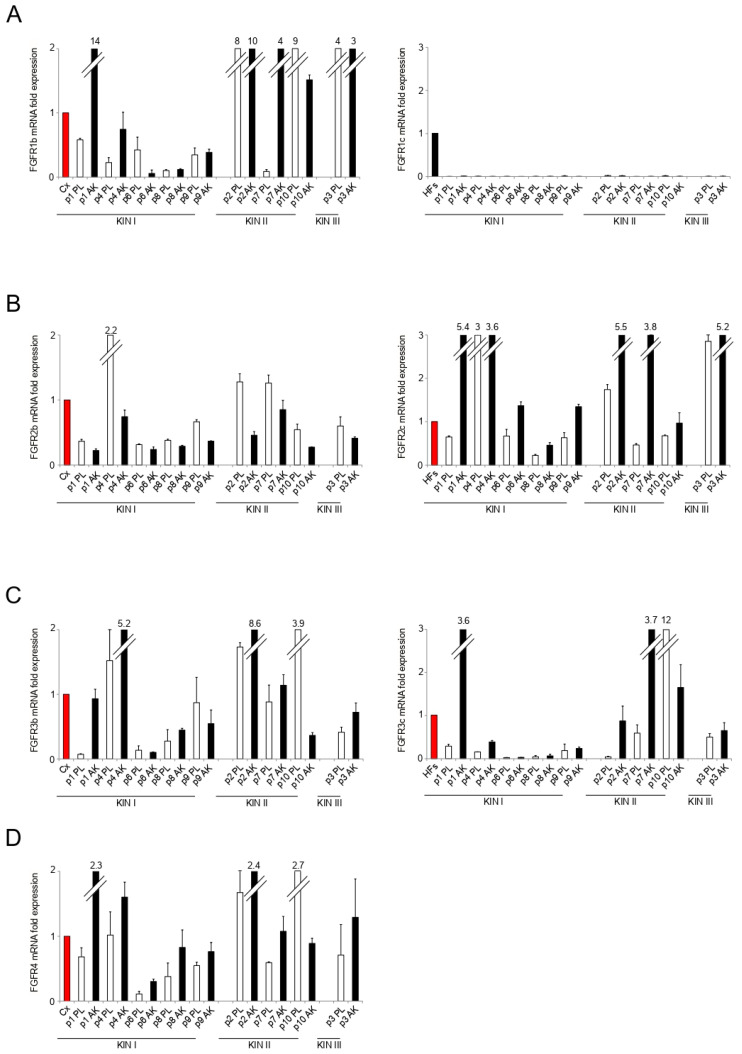
Fibroblast Growth Factor Receptors (FGFR) expression profile in the epidermal portion of KIN I, KIN II, and KIN III samples. FGFR1b/FGFR1c (**A**), FGFR2b/FGFR2c (**B**), FGFR3b/FGFR3c (**C**) and FGFR4 (**D**) mRNA levels were evaluated by real-time RT-PCR (PL: perilesional samples, white bars; AK: lesional samples, black bars) and normalized the value of primary culture of Human Fibroblasts (HFs) isolated from a dermal perilesional sample (Cx, red bar). Results are expressed as mean ± standard error (SE). For FGFR2b, FGFR2c, and FGFR4, statistical analysis of relative expression rates of KIN I AK versus KINI PL samples (**E**) or of KIN I + KIN II + KIN III AK versus KIN I + KIN II + KIN III PL samples (**F**) were performed using Wilcoxon rank-sum test and significance levels were defined as *p* < 0.05: * *p* < 0.05, ** *p* < 0.001 vs. the corresponding PL samples.

**Figure 4 biology-10-00331-f004:**
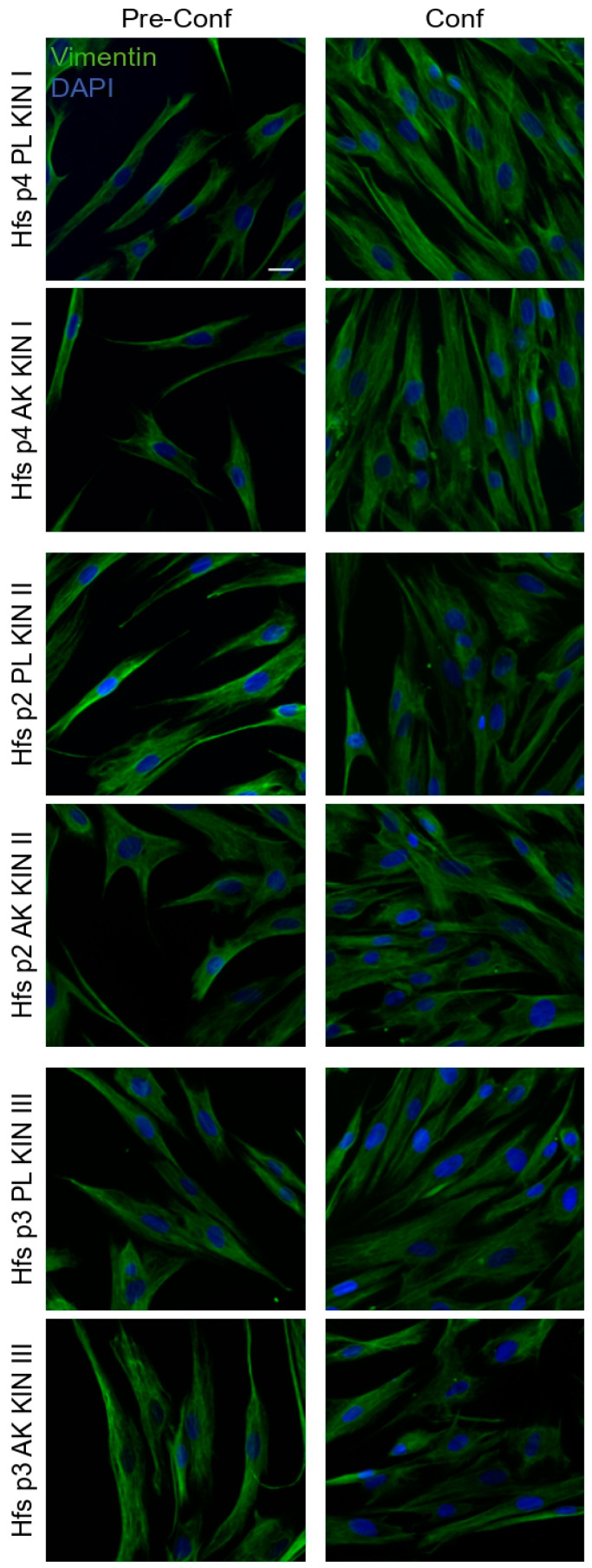
Characterization of the primary cultures of AK-associated and perilesional fibroblasts. Immunofluorescence analysis of expression of the mesenchymal marker vimentin on representative example of different cultures of human fibroblasts isolated from the dermal portion of lesional AK KIN I, KIN II, and KIN III samples and from perilesional tissues (PL). In all cultures, cells are highly positive for vimentin staining, which appears as cytoplasmic bundles of filaments. Nuclei are stained with 4′,6-diamidino-2-phenylindole (DAPI). Bar 10 µm.

**Figure 5 biology-10-00331-f005:**
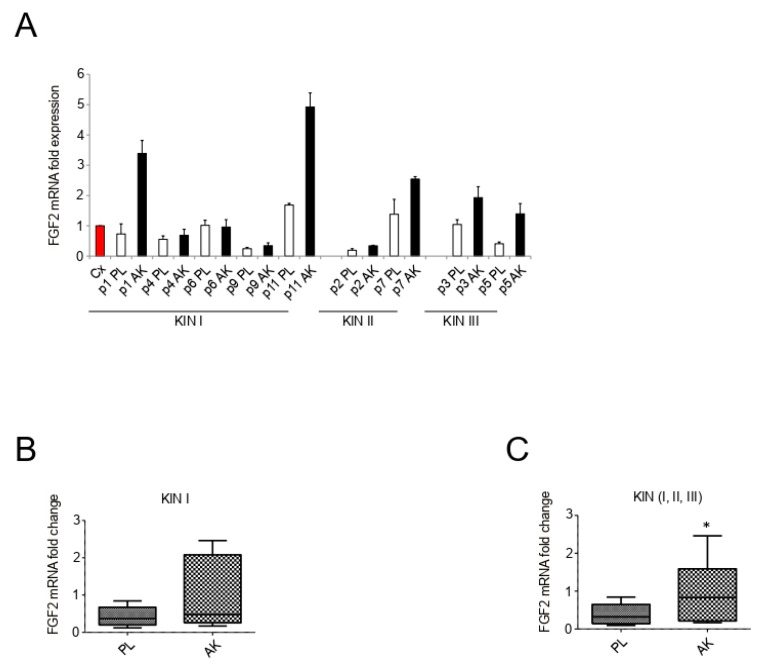
Expression of FGF2 in the dermal portion of KIN I, KIN II, and KIN III samples. FGF2 (**A**) mRNA levels were evaluated by real-time RT-PCR (PL: perilesional samples, white bars; AK: lesional samples, black bars) and normalized with respect to the mean of all the perilesional values (Cx, red bar). Results are expressed as mean ± standard error (SE). Statistical analysis of relative expression rates of KIN I AK versus KINI PL samples (**B**) or KIN I + KIN II + KIN III AK versus KIN I + KIN II + KIN III PL samples (**C**) was performed using Wilcoxon rank-sum test and significance levels were defined as *p* < 0.05: * *p* < 0.05 vs. the corresponding PL samples.

## Data Availability

All the data is provided in the article.

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
