# Peer review of "Expression Profile of Fibroblast Growth Factor Receptors, Keratinocyte Differentiation Markers, and Epithelial Mesenchymal Transition-Related Genes in Actinic Keratosis: A Possible Predictive Factor for Malignant Progression?"

_biology, 2021, doi:10.3390/biology10040331_

Round 1

Reviewer 1 Report

The authors are presenting their research regarding Transcript analysis and mRNA expression profiles for FGFRs, keratinocyte differentiation 2 markers and EMT-related genes in Actinic keratosis. The authors aim to elucidate genes that could be used as mRNA expression markers malignant progression. The topic is relevant and of interest. The article is well presented, however I have some critical points that I feel needs to be corrected before this manuscript can be accepted for publication.

Below are my specific comments and suggestions for improvement.

Page 1 row 12 and throughout the manuscript

When using abbreviations always write them out in full the first time they are used together with the used abbreviation.

Page 1 row 13; I suggest to use “investigated” or “analysed” instead of “assessed”.

Page 1 row 26; I suggest to use “may indicate” instead of “encouraged us to speculate”.

Page 2, row 101 and 103; I suggest to use “antisense” instead of “anti-sense”.

Page 2 row 125 and 129, also throughout manuscript; always specify name of manufacturer, city, country as for e.g. Invitrogen (row 125) and Bio-Rad (row 129).

Page 6 row 196; I suggest to use “transcript profile” or “gene expression profile” instead of “genetic profile”.

Page 7 row 236; I suggest to write “the FGFR expression” instead of “ FGFR expression”.

Page 7 row 237; I suggest to write “the FGFR2 isoform” instead of “FGFR2 isoform”.

Page 10 row 274-274; I suggest to write “In concert with the potential impairment of FGFR expression in keratinocytes, we therefore wanted to further investigate whether a possible deregulation of..” instead of “Then we wondered if, in concert with the impairment of FGFR expression in keratinocytes, a possible deregulation of..”.

Page 10 row 277; I suggest to write “In order to further explore this…” instead of “To assess it…”.

Throughout the manuscript; make sure that all hyphenations in the end of the rows are removed, e.g. row 50, 51, 52, 56, 57, 61, 71, 75, 76, 77, 78, 79, 84, 94, 95, 96, 102, 125, 127, 135, 136, 144, 145, 149, 157, 158, 164, 168, 170, 171, 174, 177, 183, 197, 200, 204, 206, 207, 208, 209, 210, 213, 214, 217, 221, 222, 235, 242, 243, 245, 247, 248, 249, 250, 252, 278, 282, 283, 284, 286, 287, 288, 291, 294, 296, 298, 314, 318, 327, 328, 336, 341, 344, 348, 349, 353, 354, 395, 363, 364, 375, 379, 381, 396 and figures.

Author Response

Dear reviewer #1,

Your comments of the have been very helpful and we have prepared a revised version of our manuscript following their suggestions as follows:

  • As suggested, in Page:1 row: 12 (Page:1 row: 26 of the revised version of the manuscript) and throughout all the manuscript, all the abbreviations have been reported in full, together with the used abbreviation, the first time they have been used.

  • Page:1 row: 13 (Page:1 row: 28 of the revised version) the suggested correction has been made.

  • Page: 1 row 26 (Page:1 row: 42 of the revised version) the suggested correction has been made.

  • Page: 2, row 101 and 103 (Page 3, row 131 and 133 of the revised version) the suggested correction has been made.

  • As required, throughout all the manuscript name of manufacturers, city, country have been reported.

  • Page: 6, row 196 (Page: 7, row: 245 of the revised version) the suggested correction has been made.

  • Page: 7, row 236 (Page: 9, row: 285 of the revised version) the suggested correction has been made.

  • Page: 7, row 237 (Page: 9, row: 286 of the revised version) the suggested correction has been made.

  • Page: 10, row: 274 (Page: 11, row: 324 of the revised version) the suggested correction has been made.

  • Page: 10, row: 277 (Page: 11, row: 327 of the revised version) the suggested correction has been made.

  • As suggested all hyphenations in the end of the rows throughout the manuscript have been removed.

We hope that this revised version is now suitable for publication. Thank you for your kind attention. Looking forward to hear from you soon.

Sincerely,

Francesca Belleudi PhD

Dip. Medicina Clinica e Molecolare

Sapienza Università di Roma

Viale Regina Elena 324, 00161 Roma, Italy

Phone and fax: +39-06-33775257

e-mail: francesca.belleudi@uniroma1.it

Reviewer 2 Report

Persechino et all suggests the importance of the expression of FGFRs, as well as of keratinocyte differentiation and EMT-related markers in differentially graded AK lesions as predictive molecular markers for skin cancer progression. The findings are relevant for early treatment, however the authors should provide additional results to confirm their findings. 

  • In addition to mRNA levels presented in KIN I-III, how about the protein levels of these EMT markers? Particularly, E-cadherin is lost at the cell membrane when cancer cells undergone EMT could be shown by qPCR alone.
  • How did the authors isolate the keratinocytes from AK patients? Are results presented in Figure 1 to 3 total skin? If yes, then how do the authors exclude other stromal cells? The authors should also prove they were actually keratinocytes and no contamination of other dermal cells.
  • Same with fibroblasts, the authors should also stain the fibroblasts from AK with FGF2 antibodies?
  • How about other stromal cells? Are they also affected in different stages of AK (KIN I-III)? Why only focuses on fibroblasts?

In general, the manuscript is lacking solid results to prove their hypothesis. Therefore, the manuscript could only be considered for a publication after a major revision.

Author Response

Dear reviewer #2,

Your comments of the have been very helpful and we have prepared a revised version of our manuscript following their suggestions as follows:

  • Unfortunately, the material from each biopsy was very poor and was all processed for RNA extraction; no further material for protein extraction advanced.

  • We did not isolate primary cultures of keratinocytes from our samples; we performed an efficient separation of epidermal and dermal portions using a widely-set protocol (Nanni, Ranieri et al., Cells, 2019; Rosato et al., Cell Death Dis., 2018; Nanni et al., J.Cell. Mol. Med., 2018; Ranieri et al., Mol. Carcinog., 2018) which involves the digestion of the basal membrane by a dispase solution (see Materials and Methods section). Therefore, results presented in Figure 1 to 3 are obtained using RNA extracts from the epidermal portion (and not from total skin) of each sample, as better specified in the Results section (Page: 4, row:193; Page: 7, row: 243; Page: 9, row: 290 of the revised manuscript).

  • Same goes for Figure 5; the presented results are obtained using RNA extracts from the dermal portion of each sample, as specified in the results section (Page: 11, row: 346 of the revised manuscript);

In accordance with the reviewer’s suggestion, the primary cultures of fibroblasts obtained from the dermal portions (showed in Figure 4 for Vimentin staining) were also previously checked for FGF2 staining, but we didn’t observe detectable signal in both lesional and perilesional cells. A possible explanation is that the intracellular, endogenous levels of this growth factor are too low to be appreciable by immunofluorescence approaches.

  • We focused on dermal fibroblasts because one of the main goals of our study was to investigate the dysregulation of paracrine loops between keratinocytes and fibroblasts, which could be reached via the deregulation o of FGF/FGFRs axis.

We hope that this revised version is now suitable for publication. Thank you for your kind attention. Looking forward to hear from you soon.

Sincerely,

Francesca Belleudi PhD

Dip. Medicina Clinica e Molecolare

Sapienza Università di Roma

Viale Regina Elena 324, 00161 Roma, Italy

Phone and fax: +39-06-33775257

e-mail: francesca.belleudi@uniroma1.it

Reviewer 3 Report

Authors present a quality and well-written experimental manuscript showing that expression profile of FGFRs, keratinocyte differentiation markers and EMT-related genes in AK is a possible predictive factor for malignant progression.

Authors assessed the expression pattern of FGFRs and of the ligand FGF2, as well as that of early and terminal differentiation markers and EMT-related genes in differently graded AK lesions (KIN I, KIN II; KIN III). By doing so they aimed to clarify if specific gene expression profiles could be a useful prognostic tool to identify those early AK lesions with highest probability of direct progression towards squamous cell carcinoma.

They demonstrated that the keratinocyte differentiation markers K1, DSG1 and FIL were progressively downregulated in KIN I, II and III  lesions, while the modulation of epithelial/mesenchymal markers and the induction of the transcription factors Snail1 and ZEB1 compatible with pathological EMT, even if observable, did not appear to correlate with AK progression.

Authors used a range of experimental methods, including histological samples and primary cell cultures, immunofluorescence, real-time qPCR. They observed that AK lesions display modulation of differentiation and of EMT-related markers. They also found that differentially graded AK lesions show differential modulation of FGFRs and that AK lesions show increased expression of FGF2, independently from the progression grade.

Based on the obtained results authors conclude that precocious AK lesions presenting FGFR2 isoforms switch, as well as enhanced secretion of FGF2, ingenol mebutate could act as amplifier for the PKCε-mediated oncogenic signaling favoring rapid progression of the KIN I lesions towards skin malignancies.

Other comments:

1) What method was used to analyze the PCR results? Probably Pfaffl ddCq method.

2) Would be good to have a brief conclusion and future prospect at the end of the abstract section.

3) Authors are kindly encouraged to cite the following article that describes a transcriptional factor involved in the malignant tumor progression. DOI: 10.22099/mbrc.2019.34179.1419 (Pubmed ID 31998813)

Overall, the manuscript is valuable for the scientific community and should be accepted for publication after minor corrections are made.

Author Response

Dear reviewer #3,

Your comments of the have been very helpful and we have prepared a revised version of our manuscript following their suggestions as follows:

  • As guessed by the reviewer, to analyze the PCR results, the Pfaffl ddCq method has been used, as better specified in the Materials and Methods section (Page: 4, row: 174 of the revised manuscript).

  • As suggested by the reviewer, a brief conclusion has been added at the end of the Abstract section (Page: 1 row: 45 of the revised manuscript).

  • The suggested citation has been included in the Discussion section (Page: 14, row: 441 of the revised manuscript).

We hope that this revised version is now suitable for publication. Thank you for your kind attention. Looking forward to hear from you soon.

Sincerely,

Francesca Belleudi PhD

Dip. Medicina Clinica e Molecolare

Sapienza Università di Roma

Viale Regina Elena 324, 00161 Roma, Italy

Phone and fax: +39-06-33775257

e-mail: francesca.belleudi@uniroma1.it

Reviewer 4 Report

The ms by Persechino et al needs some improvements, but it is worth of publication. In the abstact section, AK should be defined. Fig 1 and 2 needs a better quality definition, in fact, they are presented in poor quality. Discussion section should also consider possible applications of the observations.

Author Response

Dear reviewer #4,

Your comments of the have been very helpful and we have prepared a revised version of our manuscript following their suggestions as follows:

  • As suggested by the reviewer, the AK acronym has been defined in the Abstract section (Page: 1, row: 26 of the revised manuscript).

  • As requested, Figure 1 and Figure 2 with better quality definition have been inserted in the revised manuscript.

  • We agree with the Reviewer’s remark and considerations about possible applications of our findings have been included in the Discussion section (Page: 14, row: 433 of the revised manuscript).

We hope that this revised version is now suitable for publication. Thank you for your kind attention. Looking forward to hear from you soon.

Sincerely,

Francesca Belleudi PhD

Dip. Medicina Clinica e Molecolare

Sapienza Università di Roma

Viale Regina Elena 324, 00161 Roma, Italy

Phone and fax: +39-06-33775257

e-mail: francesca.belleudi@uniroma1.it

Round 2

Reviewer 2 Report

The authors have addressed my concerns and thank you for that. Although there were still some limitations, but hopefully it could be addressed in the future studies. In addition, I would like to suggest to the authors to discuss some other possible signalling cross-talk in epithelial-fibroblast cells particularly in skin cancers, in AK or SCCs (since this has been highlighted in this paper) e.g. in some recent studies DOI: 10.1016/j.celrep.2019.07.092 and https://doi.org/10.15252/emmm.201911466.

Author Response

Dear reviewer #2,

Your comments of the have been very helpful and we have prepared a revised version of our manuscript following your suggestion.

As indicated by you, the possible signaling cross-talk in epithelial-fibroblast cells particularly in skin cancers and in squamous cell carcinomas (SCCs) have been more extensively discussed in the Discussion section (Page: 14, line: 428 of the revised manuscript) and the references of some recent studies have been added in the manuscript.

We hope that this revised version is now suitable for publication. Thank you for your kind attention. Looking forward to hear from you soon.

Sincerely,

Francesca Belleudi PhD

Dip. Medicina Clinica e Molecolare

Sapienza Università di Roma

Viale Regina Elena 324, 00161 Roma, Italy

Phone and fax: +39-06-33775257

e-mail: francesca.belleudi@uniroma1.it
